# Paracrine Signaling from Breast Cancer Cells Causes Activation of ID4 Expression in Tumor-Associated Macrophages

**DOI:** 10.3390/cells9020418

**Published:** 2020-02-11

**Authors:** Sara Donzelli, Andrea Sacconi, Chiara Turco, Enzo Gallo, Elisa Milano, Ilaria Iosue, Giovanni Blandino, Francesco Fazi, Giulia Fontemaggi

**Affiliations:** 1IRCCS Regina Elena National Cancer Institute, Oncogenomic and Epigenetic Unit, Via E. Chianesi, 53, 00144 Rome, Italy; 2IRCCS Regina Elena National Cancer Institute, Pathology Department, Via E. Chianesi, 53, 00144 Rome, Italy; 3Department of Anatomical, Histological, Forensic & Orthopaedic Sciences, Section of Histology & Medical Embryology, Sapienza University of Rome, Via A. Scarpa, 16, 00161 Rome, Italy; 4Laboratory affiliated with Istituto Pasteur Italia-Fondazione Cenci Bolognetti, 00161 Rome, Italy

**Keywords:** Tumor-associated-macrophages, TAMs, breast cancer, BLBC, TNBC, ID4, YAP1, CTGF, CYR61, VEGFA, ARNT

## Abstract

Background: Tumor-associated macrophages (TAMs) constitute a major portion of the leukocyte infiltrate found in breast cancer (BC). BC cells may reprogram TAMs in a pro-angiogenic and immunosuppressive sense. We previously showed that high expression of the ID4 protein in triple-negative BC cells leads to the induction of a proangiogenic program in TAMs also through the downregulation of miR-107. Here, we investigated the expression and function of the ID4 protein in TAMs. Methods: Human macrophages obtained from peripheral blood-derived monocytes (PBDM) and mouse RAW264.7 cells were used as macrophage experimental systems. ID4-correlated mRNAs of the TCGA and E-GEOD-18295 datasets were analyzed. Results: We observed that BC cells determine a paracrine induction of ID4 expression and activation of the ID4 promoter in neighboring macrophages. Interestingly, ID4 expression is higher in macrophages associated with invasive tumor cells compared to general TAMs, and ID4-correlated mRNAs are involved in various pathways that were previously reported as relevant for TAM functions. Selective depletion of ID4 expression in macrophages enabled validation of the ability of ID4 to control the expression of YAP1 and of its downstream targets CTGF and CYR61. Conclusion: Collectively, our results show that activation of ID4 expression in TAMs is observed as a consequence of BC cell paracrine activity and could participate in macrophage reprogramming in BC.

## 1. Introduction

Macrophages, crucial players in the adaptive immune response, are usually involved in detection and destruction of pathogens and apoptotic cells; however, when recruited by breast cancer (BC) cells in the tumor stroma, macrophages actively participate in tumor progression. Specifically, tumor-associated macrophages (TAMs) are “educated”, through paracrine signaling from BC cells, to exert pro-angiogenic and immunosuppressive functions [1,2,3]. Several reports suggest that TAMs adopt a trophic immunosuppressive phenotype that is functionally reminiscent of the alternatively activated type II (M2) macrophages [4]. However, TAMs present great phenotypic diversity depending on the combinations of stimuli received in the tumor microenvironment, and then multiple subpopulations of TAMs might exist within a tumor, which probably change temporally during cancer progression and geographically on the basis of their location within the tumor [5,6].

The helix-loop-helix ID4 protein is a member of the ID (Inhibitors of Differentiation) family of proteins (ID-1 to ID-4) that act as dominant-negative regulators of basic helix-loop-helix transcription factors [7]. ID4 is a negative regulator of several key pathways involved in luminal fate specification in the mammary gland [8]. In the context of breast cancer, high ID4 expression is associated with triple-negative breast cancer (TNBC) and basal-like breast cancer (BLBC) subtypes, where the ID4 gene might be amplified/overexpressed, and marks a subset of cancers with poor prognosis [8,9,10]. Functionally, it has been shown that ID4 may exert similar inhibitory effects on *BRCA1* and estrogen receptor alpha (*ERα*) gene expression, and, conversely, BRCA1 and ERα may demonstrate redundancy in inhibiting *ID4* gene expression in breast cancer cells and tissues [8]. Moreover, ID4 enhances the angiogenic potential of breast cancer cells through the post-transcriptional regulation of IL8, CXCL1, and VEGFA mRNAs and through the reprogramming of tumor-associated macrophages [11,12,13,14]. High expression of ID4 in BC cells indeed enhances macrophage motility and leads to the activation of a pro-angiogenic program in TAMs, which involves both the transcriptional increase of angiogenic factors, such as granulin (GRN), and the downregulation of antiangiogenic miR-15/107 group members (e.g., miR-107, miR-15b, and miR-195) [12]. Accordingly, ID4 mRNA levels robustly predict survival, specifically in the subset of tumors showing high macrophage infiltration [12]. The chromosomal region containing ID4 (6p22) is amplified in 32% of high-grade serous ovarian cancers (HG-SOC) [15], and ID4 is over-expressed in most primary ovarian cancers and ovarian cancer cell lines, but not in normal ovaries [16]. In HG-SOC, inhibition of ID4 in vivo suppresses the growth of established tumors and significantly improves survival, suggesting that targeting ID4 expression is a viable therapeutic strategy in cancers that over-express ID4 [16].

In this study, starting from the observation that breast cancer cells induce the expression of ID4 in neighboring macrophages, we explored the mechanisms of ID4 activation and the functional involvement of ID4 in TAM activity.

## 2. Materials and Methods

### 2.1. Cell Cultures and Transfections

Breast cancer cell lines SKBR3 (kindly provided by M. Oren’s lab, Weizmann Institute of Science, Rehovot, Israel) MDA-MB-468 (ATCC), HCC-1954 (ATCC), and monocytic cell lines HL60, U937, Monomac-3, and THP1 were grown at 37 °C with 5% CO_2_ and maintained in RPMI medium (Invitrogen-GIBCO, Carlsbad, CA, USA), containing 10% heat-inactivated (HI) foetal bovine serum (FBS) (Invitrogen-GIBCO) and penicillin/streptomycin. The OVCAR3 (ATCC) cell line was cultured as described above in RPMI (Invitrogen-GIBCO) supplemented with 20% HI-FBS and 0.01mg/mL insulin. The RAW264.7 cell line was cultured as described above in DMEM medium (Invitrogen-GIBCO) 10% HI-FBS (Invitrogen-GIBCO). HL60 and U937 cells were differentiated by treatment with 1,25-dihydroxyvitamin D3 (VitD3) (Sigma–Aldrich, St. Louis, MO, USA) at a concentration of 250 ng/mL for 72h. Monocytic differentiation was assessed by fluorescence-activated cell sorting (FACS) as previously reported [12] using allophycocyanin (APC) anti-human CD11b (BD Biosciences, San Jose, CA, USA), PerCP-Cy5.5 anti-human CD14 (BD Biosciences), and the PE-IgG1 isotype control (eBiosciences Inc., San Diego, CA, USA) antibodies for the evaluation of CD11b–CD14 co-expression as a marker of monocytic differentiation. A minimum of 10,000 events was collected for each sample with a flow cytometer (CyAN ADP, Dako, Glostrup, Denmark) using Summit 4.3 software (Beckman Coulter, Fullerton, CA, USA) for data acquisition and analysis.

An expression vector containing the HA-tagged ID4 coding sequence [17] or control empty vector was transfected in cancer cells using Lipofectamine 2000 reagent (Thermo Fisher Scientific, Waltham, MA, USA) in ID4-overexpression experiments. RNAiMax reagent (Thermo Fisher Scientific) was used to transfect siRNAs in BC cells. Sequences of siRNAs directed to ID4 were previously reported [12,13]. HL60 cells were treated with macrophage-activating compounds: LPS 1μg/mL (Sigma–Aldrich) or TNF-alpha 50 ng/mL (Sigma–Aldrich) or IL4/IL13 20ng/mL (BD Biosciences).

Human peripheral blood-derived monocytes (PBDM) were isolated from blood donors using Lymphoprep solution (Axis-Shield, Dundee, UK) followed by the isolation of CD14+ cells with the Monocyte Isolation Kit II (Miltenyi Biotec, Bergisch Gladbach, Germany). Differentiation was achieved through 1-week culturing in RPMI medium containing recombinant CSF1 (hMCSF, Cell Signaling Technology, Danvers, MA, USA, #8929SC). Macrophages were transfected with siRNAs directed to ID4 mRNA using the TransIT-X2^®^ Dynamic Delivery System (Mirus, Madison, WI, USA) following the manufacturer’s instructions and were collected after 48 h.

Conditioned media (CM) from BC and OVCAR3 cells were prepared by culturing cells for 24 h in serum-free RPMI medium. CM were centrifuged to eliminate cell residues before preparation of aliquots and storage at −80 °C. When si-ID4 BC cells were used to prepare CM, we always collected CM before 48h from transfection; the proliferation of cells was delayed after this time point under the si-ID4 condition.

### 2.2. Immunohistochemistry

Collection of tumors from BC patients was reviewed and approved by the ethics committee of the Regina Elena National Cancer Institute (IFO1270/19) and contained data for which written informed consent was obtained from all patients. BC specimens for IHC analysis were fixed for 18–24 h in 4% (*v*/*v*) buffered formaldehyde and were then processed with paraffin wax. ID4 and CD68 proteins were evaluated by IHC in 5-μm-thick paraffin-embedded tissues using anti-ID4 (MAB4393, EMD Millipore, Billerica, MA, USA) and anti-CD68 (LEICA clone 514H12) antibodies. Immunoreactions were revealed by a streptavidin–biotin-enhanced immunoperoxidase technique (Super Sensitive MultiLink) in an autostainer (Bond III, Leica Biosystems, Wetzlar, Germany) [18]. Diaminobenzidine (DAB) was used as a chromogenic substrate.

### 2.3. Tissue Immunofluorescence

Tissue sections were deparaffinised, rehydrated, and heated in Antigen Retrieval Citra Solution, pH 6. Then, sections were incubated for 5 min with 0.25% Triton in PBS 1%. Sections were blocked for non-specific binding with PBS–BSA 5% for 1 h and were incubated with primary antibodies, anti-ID4 (B-5, Santa Cruz Biotechnology, Dallas, TX, USA) and anti-CD163 (D6U19, Cell Signaling), in PBS–BSA 1% for 16 h at 4 °C. The following day, cells were washed three times with PBS 1% followed by incubation with Alexa Flour 488 (rabbit) and Alexa Fluor 594 (mouse)-conjugated secondary antibodies (Molecular Probes Inc., Eugene, OR, USA) for 2 h at RT. After washing three times with PBS, sections were counterstained with DAPI for 5 min and were mounted with Vectashield (Vector Labs, Burlingame, CA, USA). Cells were examined under a Zeiss LSM 510 laser scanning fluorescence confocal microscope (Zeiss, Wetzlar, Germany). 

### 2.4. Immunocytochemistry

For immunocytochemistry assay, SKBR3 and MDA-MB-468 cells were seeded onto glass coverslips (Marienfeld, Lauda-Königshofen, Germany) in 6-well dishes (Corning Inc., NY, USA) at 4 × 10^4^ cells/well and were transfected with siSCR, siID4#1 or siID4#2, while differentiated HL60 cells grown in the presence of RPMI or CM (48 h) were concentrated onto microscope slides using cytospin. Cells were fixed with 4% formaldehyde in PBS for 15 min at RT and were then permeabilized with 0.25% Triton X-100 in PBS for 10 min. After washing with PBS, the cells were incubated with peroxidase inhibitor and then with anti-ID4 antibody (H70, Santa Cruz Biotechnology, Dallas, TX, USA) diluted 1:200 in 5% bovine serum albumin (BSA)/PBS for 2 h at RT. Protein staining was revealed through a DAB enzymatic reaction, while nuclei were counterstained with haematoxylin.

### 2.5. Western Blotting and Antibodies

For the Western blot analysis, cells were lysed in RIPA buffer or 8 M urea. The protein concentration was measured using the Bio-Rad Protein Assay Kit (Hercules, CA, USA). The lysate was mixed with 4× Laemmli buffer. Total protein extracts were resolved on polyacrylamide gel and were then transferred onto a nitrocellulose membrane. The following primary antibodies were used: Gapdh (sc-32233), ID4 (H70) sc-13047, ID4 (B5) sc-365656, and HA (12CA5) sc-57592 (Santa Cruz Biotechnology, Dallas, TX, USA), ARNT (TA501147, Origene). A secondary antibody fused with horseradish peroxidase (HRP) was used for chemiluminescence detection on a UVITEC instrument. In the VEGFA blocking experiments, antibody (anti-VEGFA, AF-293-NA; R&D Systems, Minneapolis, MN, USA) was added to CM and incubated for 30 min at RT before being used to culture macrophages, following the manufacturer’s instructions.

### 2.6. RNA Isolation

#### 2.6.1. RT-qPCR and PCR

RNA was isolated with Trizol (Invitrogen, Carlsbad, CA, USA), and its concentration was measured using a NanoDrop 2000 (Nanodrop Technologies, Wilmington, DE, USA). Reverse transcription was performed with M-MLV Reverse Transcriptase (Thermo Fisher Scientific). qPCR was carried out on an ABI PRISM 7500 Fast Sequence Detection System (Applied Biosystems, Carlsbad, CA, USA). Primers used for PCR analyses are available upon request. The expression values of mRNAs were calculated by the ΔΔCt method and were normalized with housekeeping control genes (GAPDH, β-actin, H3, Ald-A).

#### 2.6.2. Formaldehyde Cross-Linking and Chromatin Immunoprecipitation

Formaldehyde cross-linking and chromatin immunoprecipitations were performed as previously described [11]. Briefly, RAW264.7 or U937 cells were crosslinked using 1% formaldehyde/PBS for 10 min at RT. Chromatin from one 150 mm dish (50% confluence) was used for each immunoprecipitated Ab. Recovery of immune complexes was performed using Dynabeads^®^ protein G (Invitrogen, Carlsbad, CA, USA). The chromatin solution was immunoprecipitated with anti-ARNT (Anti-HIF1 beta antibody-ChIP Grade ab2, abcam, Cambridge, UK), anti-H4Ac (cell signaling) or anti-H3K9me3 and H3K9Ac (cell signaling). The mouse ID4 promoter was evaluated by real-time PCR using the following primers: for-CCATCCTGGCCCGACTCCCA, rev-GCCACCTCGGGGAATGACGC. Enrichments were normalized with respect to the GAPDH promoter. The human ID4 promoter was analyzed using primers for regions A and B that were previously reported [11].

#### 2.6.3. Bioinformatic Analysis and Statistics

The association between gene and miRNA normalized expression was assessed by calculating the Spearman correlation coefficient. P-values were calculated using a paired, two-tailed T test, where indicated in the figure legend, using Prism 7.00 (GraphPad Software, La Jolla California, CA, USA). 

## 3. Results

### 3.1. ID4 Expression in Breast Cancer Cells Causes ID4 Induction in TAMs

We recently reported that ID4 expression correlates with macrophage recruitment in triple-negative breast cancer (TNBC) [12]. Within this study, we observed that TNBC tissues showing high macrophage infiltration, evaluated using CD68 as a macrophage marker, presented ID4 protein expression not only in cancer cells but also in cells of the leukocyte infiltrate (Appendix A). This prompted us to evaluate whether the ID4 protein is expressed in macrophages infiltrating BC tissues. To this end, we performed double immunofluorescence analysis using antibodies recognizing the ID4 protein and the macrophage marker CD163. As shown in Figure 1A, we observed that a large part of the macrophage population expresses the ID4 protein in the tumor stroma. We next analyzed whether the ID4 expression level changes in macrophages cultured in conditioned medium (CM) from breast cancer cells. As shown in Figure 1B, ID4 mRNA is induced in human peripheral blood-derived macrophages (PBDM) grown for 48h with CM from the BC cell lines MDA-MB-468, HCC1954, and SKBR3. A similar induction is also observed with CM from ovarian cancer cells OVCAR3 (Figure 1B). Using additional macrophage cultures, such as those derived from U937 and HL60 cell lines, we confirmed that ID4 is induced at mRNA and protein levels in the presence of CM from SKBR3 or MDA-MB-468 cells (Figure 1C–E and Appendix A). Interestingly, additional well-characterized macrophage-activating stimuli, such as LPS, TNF-alpha, and IL-4/IL-13, did not modulate ID4 expression, indicating that this effect is not generically associated with macrophage activation but is specifically obtained as a paracrine effect of BC cell activity (Appendix A). Moreover, the culture of human fibroblasts, another cell type usually present in the tumor stroma, with CM from breast cancer cells did not lead to ID4 induction (Appendix A), indicating that the observed effect is specific to the monocyte/macrophage lineage. 

Interestingly, co-culture of macrophages with breast cancer cells whose ID4 expression has been inhibited by siRNA transfection (si-ID4#1, si-ID4#2) led to a significantly attenuated ID4 induction in macrophages, compared to co-culture with control cells (si-SCR) (Figure 2A–D). On the contrary, PBDM cultured with conditioned medium (CM) derived from ID4-overexpressing breast cancer cells (ID4-HA) showed a much stronger and time-dependent ID4 mRNA induction compared to the control CM (empty vector, EV) (Figure 2E). ID4 overexpression effects were recapitulated in various additional macrophage experimental systems (Figure 2F–G and Appendix A). Together, these results demonstrate that breast cancer cells determine a paracrine induction of ID4 expression in macrophages. This induction is strictly dependent on the ID4 expression levels in breast cancer cells.

### 3.2. ID4 Induction in Macrophages Depends on ID4 Promoter Activation

As we observed that ID4 expression in macrophages is strongly related to ID4 levels in BC cells, we wondered whether ID4 induction in macrophages depended on activation of the macrophage endogenous ID4 promoter or on the transfer of the ID4 mRNA/protein from cancer cells to macrophages. To address this, we first over-expressed HA-tagged ID4 protein in cancer cells, collected the CM and evaluated whether the ID4-HA protein was transferred to macrophages cultured with this CM. To this end, we used an antibody recognizing the HA-tag, present only on the protein exogenously expressed in cancer cells. As shown in Figure 3A–B, we did not detect the ID4-HA protein in macrophages, despite that ID4 mRNA was efficiently induced, indicating that the ID4 protein was not transferred from cancer cells to macrophages. Next, we cultured the mouse macrophage cell line RAW264.7 with CM from human BC cells and we evaluated whether the human or murine ID4 mRNA was detectable in mouse macrophages. As shown in Figure 3C, primers recognizing mouse ID4 mRNA were able to detect ID4 induction in macrophages, while human-specific primers did not detect any ID4 mRNA in RAW264.7 cells. Together, these results indicate that no ID4 protein or mRNA transfer occurred from cancer cells to macrophages. 

We then investigated the mechanism leading to ID4 expression activation in macrophages. Previous studies from our group showed that the transcriptional regulator ARNT (also known as HIF1-beta) is controlled by ID4-dependent microRNAs in macrophages [12]. Interestingly, we also observed that ARNT behaves similarly to ID4 mRNA in macrophages cultured in the presence of CM from control (si-SCR) or ID4-depleted (si-ID4) BC cells (Figure 3D). Moreover, as we previously observed that the ID4 protein controls VEGFA expression in breast cancer cells [13], and VEGFA in turn participates in the reprogramming of macrophages in a pro-angiogenic sense [12], we decided to investigate the impact of VEGFA on ID4 and ARNT expression in macrophages. As shown in Figure 3E, addition of the VEGFA blocking antibody to the CM used to culture macrophages impaired the induction of both ID4 and ARNT compared to control IgG. On the basis of these results, we explored whether ARNT could participate in ID4 transcriptional regulation in macrophages. To address this question, we analyzed the ID4 promoter sequence and observed the existence of various AhR/ARNT binding sites. We then evaluated whether ARNT was recruited on the ID4 promoter in macrophages cultured in CM from MDA-MB-468 BC cells. ChIP experiments highlighted that the ARNT protein was indeed recruited on the two analyzed regions of the ID4 promoter, enclosing AhR/ARNT consensus sequences, specifically in macrophages cultured with CM (Figure 3F–G). ChIP analysis of acetylated histones showed an increased acetylation degree only in region B of the ID4 promoter (Figure 3H–I).

### 3.3. Identification of ID4-Associated Pathways in TAMs

To investigate the functional relevance of ID4 expression in TAMs, we next evaluated publicly available datasets. A study by Ojalvo S and colleagues [19] included gene expression analysis of macrophages associated with invasive tumor cells compared to general TAMs from PyMT mammary tumors (E-GEOD-18295). In this dataset, we interestingly observed that the ID4 mRNA level was significantly higher in invasive vs. general TAMs (Figure 4A). A similar behavior was observed for CXCL1, previously identified, together with IL8, as targeted by ID4 in BC cells [11] (Figure 4A). To identify genes correlated with ID4 in macrophages and relevant to human BC, we extracted all the ID4-correlated mRNAs in the E-GEOD-18295 dataset and, subsequently, we intersected this list with that of ID4-correlated genes in basal-like BC (BLBC) from the TCGA study. BLBC was chosen, as ID4 is specifically upregulated and associated with decreased metastasis-free survival in this subtype [8,9,10]. We identified 268 and 84 mRNAs that were positively and negatively correlated with ID4, respectively, in both datasets. Interestingly, we noticed that many components of the extracellular matrix (ECM) were positively correlated with ID4 (such as collagens *COL2A1, COL4A1, COL4A2, COL5A2, COL8A1, COL9A2, COL9A3* and laminins *LAMA4, LAMB1*), suggesting that ID4 might contribute to ECM stiffening in BLBC. To identify enriched pathways among genes positively correlated with ID4, we interrogated ConsensusPathDB software, thus highlighting that ID4 associates with various pathways related to the interaction between membrane receptors and the extracellular matrix (ECM-receptor interaction) as well as with cell motility (focal adhesion, Amoebiasis) (Table 1). 

### 3.4. ID4 Controls the Expression of Hippo Pathway Members YAP1, CTGF, and CYR61 in TAMs

Of note, our bioinformatic analysis revealed that YAP1, a member of the Hippo pathway, and its downstream *bona fide* transcriptional targets, encoding the matricellular proteins CTGF and CYR61, are positively correlated with ID4 in macrophages. This particularly caught our attention given that the YAP1-dependent cascade is an important player in tissue stiffness. Correlation between ID4 and YAP1, ID4 and CTGF, ID4 and CYR61 in the E-GEOD-18295 and TCGA-basal datasets is shown in Figure 5A–B. Association between ID4 and Hippo pathway members might rely on the fact that they are commonly regulated by an upstream regulatory factor or that they are interdependent. To address this, we explored whether ID4 controls the expression of ECM remodeling factors YAP1, CTGF, and CYR61 in macrophages by depleting ID4 expression in PBDM-derived macrophages. As shown in Figure 5C, expression of YAP1, CTGF, and CYR61 mRNAs decreased in macrophages depleted of ID4 expression by siRNA transfection, indicating that ID4 controls the expression of these genes, while no modulation of macrophage receptor CCR2 was observed. 

## 4. Discussion

In this study, we show that expression of the ID4 protein is induced in macrophages cultured with conditioned growth medium from breast and ovarian cancer cells. Interestingly, efficient induction of ID4 in macrophages is obtained only if high levels of ID4 are present in breast cancer cells. This observation fits well into the context of our previous study that highlighted how high ID4 expression in BC cells causes induction of proangiogenic factors in neighboring macrophages [12]. ID4 induction could then be included in the reprogramming of macrophages caused by ID4 expression in BC cells, also involving the activation of other factors such as granulin (GRN), ephrin B2 (EphB2), and neuropilin-2 (NRP2) [12]. We also investigated the mechanisms leading to ID4 induction in macrophages. Specifically, we observed that the VEGFA present in conditioned medium is required for ID4 induction in macrophages. We previously reported that VEGFA expression is modulated by ID4 in cancer cells [13]. VEGFA then acts as the mediator that confers to ID4 protein in cancer cells the ability to cause ID4 induction in macrophages. However, we observed that depletion of ID4 in cancer cells only partially impairs the induction of ID4 in macrophages. This could rely on the fact that VEGFA expression in cancer cells is strictly regulated and depends not only on ID4 but also on several other regulatory pathways, and it is possible that ID4 depletion is not sufficient to completely abolish VEGFA activity and, consequently, ID4 expression in macrophages. 

Of note, we observed that VEGFA also controls the expression of ARNT in macrophages. ARNT (aryl hydrocarbon receptor nuclear translocator), also known as HIF1B, is a transcription factor that interacts with the ligand-bound AhR (aryl hydrocarbon receptor) protein and facilitates its translocation to the nucleus, where it modulates the expression of genes involved in xenobiotic metabolism. Recently, AhR has been shown to support BC growth by controlling reactive oxygen species (ROS) levels and the tumor-promoting features of macrophages [20]. ARNT is also a co-factor for transcriptional regulation by hypoxia-inducible factor 1 (HIF1A). Specifically, we found that ARNT expression in macrophages is, similar to ID4, decreased by VEGFA blocking in conditioned medium. As ARNT is a target of microRNA-107 (miR-107) [12,21], its downregulation in the presence of VEGFA blocking probably relies on upregulation of miR-107, which occurs in this experimental condition, as we previously reported [12]. Interestingly, we observed that the ARNT protein, which increases in macrophages in the presence of CM from BC cells, is also recruited onto ID4 promoter, containing AhR/ARNT response elements, suggesting its involvement in the transcriptional induction of ID4 occurring as a consequence of paracrine signaling from BC cells. It will be of interest to further determine whether ARNT cooperates with AhR in the control of the ID4 promoter.

To evaluate the possible functional implication of the ID4 protein in macrophages, we searched for genes and pathways correlated with ID4 in a tumor-associated macrophage dataset as well as in the TCGA dataset, the latter being necessary to identify ID4-associated factors relevant for human cancer. This was performed specifically on the basal-like subgroup of BC, as our group and others previously showed that ID4 is a powerful prognostic indicator in this subgroup [9,10,12,13]. Bioinformatic analysis enabled us to highlight some genes of the Hippo pathway that are positively associated with ID4 in these datasets. These genes are particularly interesting to us, as they have been extensively shown to play crucial roles in cancer cells, including TNBC cells, and also in immune cells of the tumor microenvironment [22,23,24]. The co-activator YAP recently emerged as a mechanosensor and a key mediator of the biological effects that are observed in response to ECM elasticity and cell shape [25,26]. Moreover, high tumor stiffness and F‑actin polymerization correlate with increased activity of YAP [27]. Interestingly, tissue stiffness is caused by collagen deposition and by the activity of lysyl oxidases [28], which increase collagen crosslinking. Importantly, tumor fibrosis has been shown to be controlled by macrophages in cancer and to play a crucial role in immune escape [29,30,31]. In the present study, we interestingly highlighted on the one hand that ID4 expression in macrophages is associated with high levels of proteins of the ECM, such as collagens and laminins, as well as of the focal adhesions, and on the other hand that the ID4 protein increases the mRNA level of YAP1 and its downstream targets CTGF and CYR61 in macrophages. As CTGF and CYR61 are matricellular proteins important for ECM-cell interaction [32], we aim to investigate in our future studies if the ID4-driven axis in macrophages contributes to the remodeling of the ECM in breast tumors and to decipher the specific roles of the ID4 protein in the ECM-Hippo signaling cascade in TAMs.

## Figures and Tables

**Figure 1 cells-09-00418-f001:**
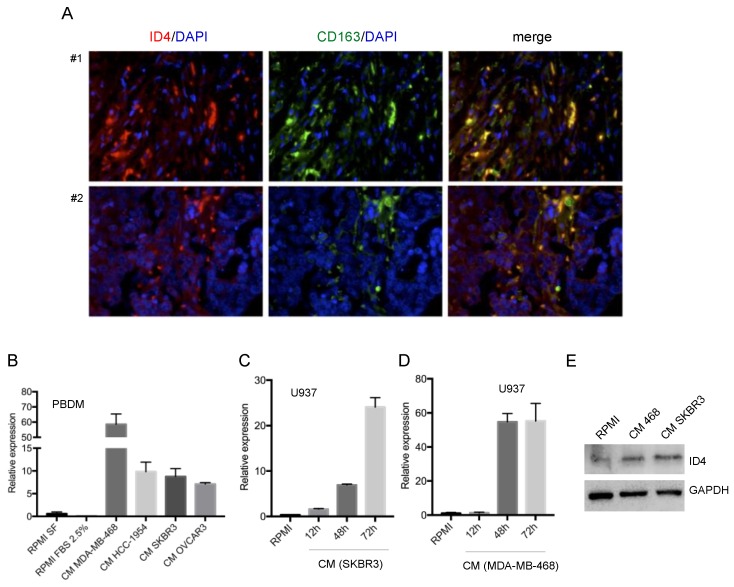
ID4 expression is induced in macrophages cultured with conditioned media (CM) from breast and ovarian cancer cells. (**A**) ID4 and CD163 protein expression evaluated by double immunofluorescence in tumor stroma (FFPE tissue sections from two breast cancer cases). (**B**) ID4 mRNA expression evaluated by RT-qPCR in macrophages, obtained from differentiation of peripheral blood-derived monocytes (PBDM), cultured with RPMI serum free (SF), RPMI 2.5% FBS or conditioned medium (CM) from the indicated breast (MDA-MB-468, HCC-1954, SKBR3) and ovarian (OVCAR3) cancer cells. Relative expression was assessed by normalizing to GAPDH levels. Data are presented as the mean of the expression observed in two donors ± standard error. (**C**,**D**) ID4 mRNA expression in differentiated U937 cells, cultured with RPMI medium or with CM from the indicated breast cancer cell lines for the indicated time. (**E**) ID4 protein expression analyzed by Western blot in differentiated U937 cells cultured in RPMI medium or in CM from MDA-MB-468 (CM 468) or SKBR3 breast cancer cell lines for 24 h.

**Figure 2 cells-09-00418-f002:**
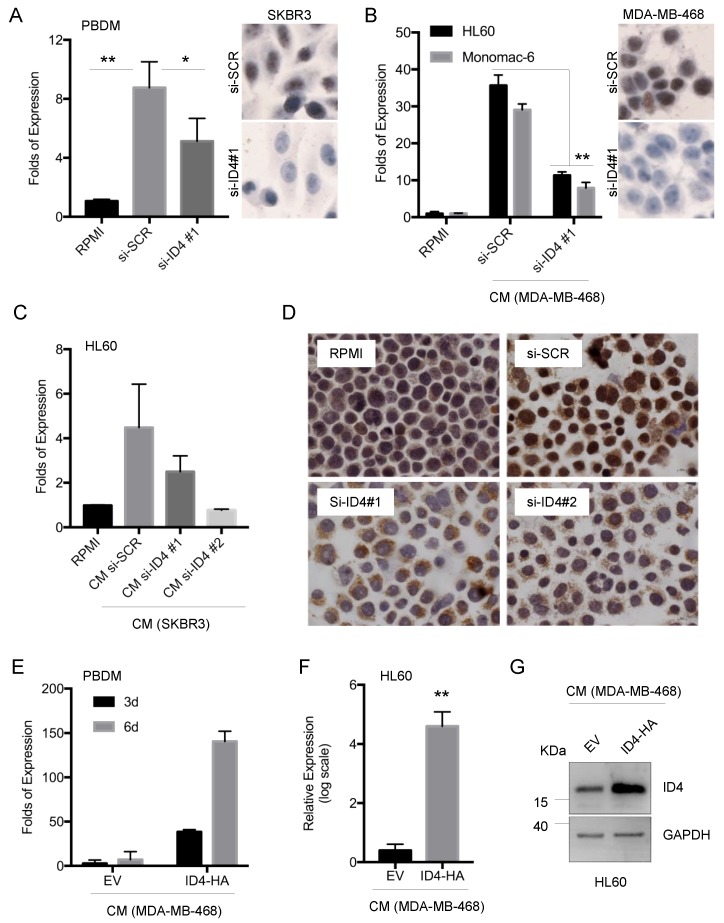
Dependency between ID4 expression levels in BC cells and TAMs. (**A**) ID4 mRNA expression in macrophages derived from PBDM co-cultured with control (si-SCR) or ID4-depleted (si-ID4) SKBR3 cells for 48 h. Right panels show ID4 protein levels, evaluated by immunocytochemistry, in control (si-SCR) and ID4-depleted (si-ID4#1) SKBR3 cells used for the co-culture. (**B**) ID4 mRNA expression in differentiated HL60 cells and in monomac-6 cells cultured with RPMI medium or with CM from the control (si-SCR) or ID4-depleted (si-ID4) MDA-MB-468 cells. Right panels show ID4 protein levels, evaluated by immunocytochemistry, in the control (si-SCR) and ID4-depleted (si-ID4#1) MDA-MB-468 cells used for the preparation of CM. (**C**,**D**) ID4 mRNA (**C**) and protein (**D**) expression evaluated by RT-qPCR and immunocytochemistry, respectively, in differentiated HL60 cells cultured with RPMI medium or with CM from control (si-SCR) or ID4-depleted (si-ID4#1 and si-ID4#2) SKBR3 cells. (**E**) ID4 mRNA expression on PBDM cultured with CM from control (EV) and ID4-HA-overexpressing MDA-MB-468 cells for the indicated number of days. F-G. ID4 mRNA (**F**) and protein (**G**) expression evaluated on macrophages obtained from HL60 differentiation and cultured in CM from control (EV) or ID4-HA-overexpressing (ID4-HA) MDA-MB-468 cells. * (*p* < 0.05), ** (*p* < 0.005). P-values were calculated by a paired two-tailed t-test.

**Figure 3 cells-09-00418-f003:**
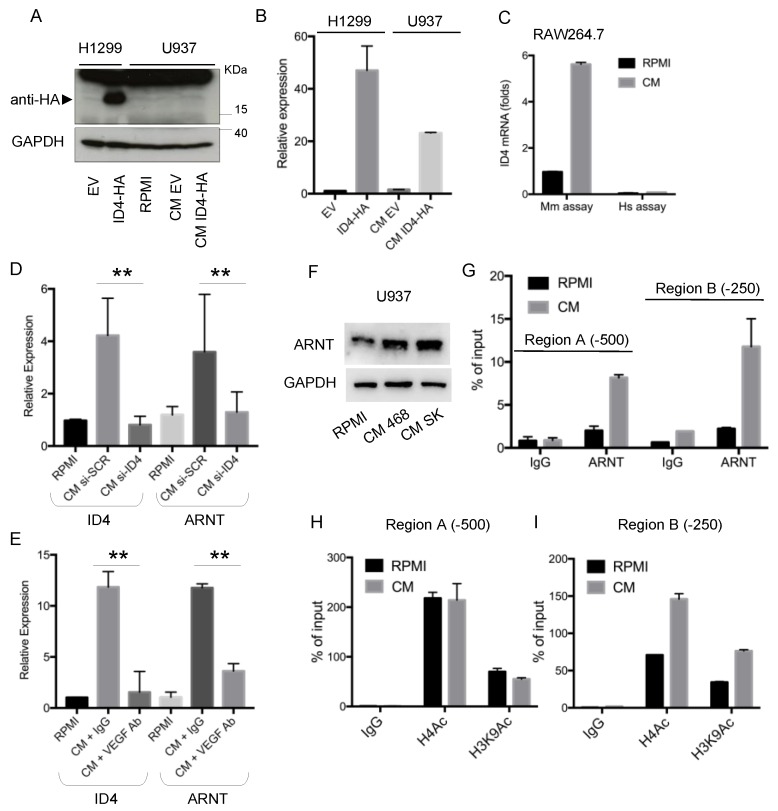
(**A**,**B**) ID4 expression analyzed by Western blot (**A**) and RT-qPCR (**B**) in recipient H1299 cells, transfected with an empty vector (EV) or an HA-tagged ID4 expression vector, and in U937 cells cultured with RPMI, CM from H1299-EV or CM from H1299-ID4-HA cells. (**C**) ID4 mRNA expression in RAW264.7 mouse macrophages grown with RPMI medium or with CM from human breast cancer cells evaluated with primers recognizing the mouse (Mm assay) or human (Hs assay) ID4 cDNA sequence. (**D**) ID4 and ARNT mRNA expression evaluated by RT-qPCR in differentiated U937-derived macrophages cultured with RPMI or with conditioned medium (CM) from control (si-SCR) or ID4-depleted (si-ID4) MDA-MB-468 breast cancer cells for 48 h. Data were normalized with respect to GAPDH expression. *p*-values were calculated on the fold decrease between si-SCR and si-ID4 conditions. (**E**) ID4 and ARNT mRNA expression evaluated by RT-qPCR in differentiated U937-derived macrophages cultured with RPMI or with conditioned medium (CM) from MDA-MB-468 breast cancer cells in the presence of the control antibody (IgG) or VEGFA blocking antibody (VEGFA Ab) for 48h. Data were normalized with respect to GAPDH expression. *p*-values were calculated on the fold decrease between IgG and VEGF Ab conditions. (F) ARNT protein expression evaluated by Western blot in differentiated U937 cells cultured with RPMI or with CM from MDA-MB-468 (CM 468) or SKBR3 (CM SK) for 24h. (**G**–**I**) Chromatin immunoprecipitation analysis of differentiated U937 cells cultured with RPMI or with conditioned medium (CM) from MDA-MB-468 breast cancer cells for 24h. IP was performed with anti-ARNT (**G**) or anti-acetylated histone H4 or anti-acetylated histone H3K9 (**H**,**I**). Two regions of the ID4 promoter located nearly at −500 (region A) and −250 (region B) bp from TSS were analysed. ** (*p* < 0.005). P-values were calculated by a paired two-tailed t-test.

**Figure 4 cells-09-00418-f004:**
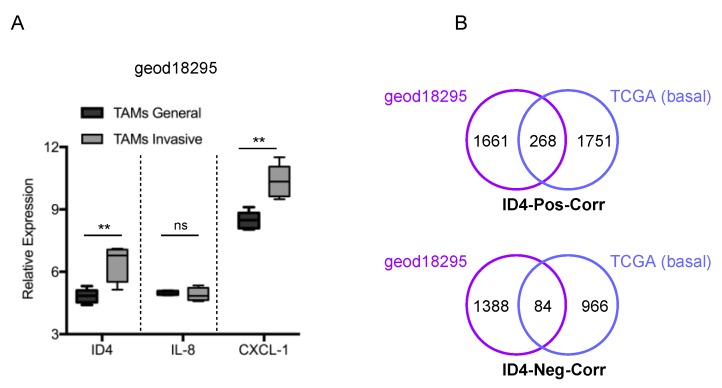
(**A**). Expression of ID4 and CXCL1 mRNA is higher in macrophages associated with invasive tumor cells (PyMT mammary tumors), compared to general TAMs, in the E-GEOD-18295 dataset. ** (*p* < 0.005) *p*-values were calculated by a two-tailed t-test. ns: not significant. (**B**). Venn diagrams showing the number of mRNAs commonly correlated to ID4 (*p* > 0.05) in the E-GEOD-18295 dataset and in basal-like tumors of the breast cancer TCGA dataset.

**Figure 5 cells-09-00418-f005:**
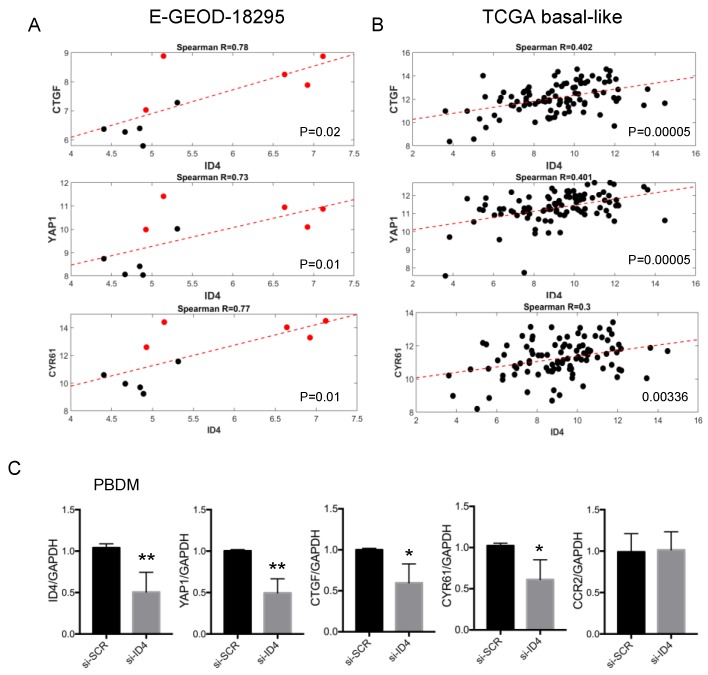
(**A**) Graph showing the correlation (Spearman coefficients) between ID4 and CTGF (upper graph), YAP1 (middle graph) or CYR61 (lower graph) in the geod18295 dataset. Black and red dots refer, respectively, to general and invasive TAMs. (**B**) Graph showing the correlation (Spearman coefficients) between ID4 and CTGF (upper graph), YAP1 (middle graph) or CYR61 (lower graph) in the TCGA dataset, considering the subgroup of basal-like breast tumors with high macrophage infiltration (CD68-high). (**C**) Expression of ID4, YAP1, CTGF, CYR61, and CCR2 in macrophages depleted of ID4 expression by siRNA delivery and in control macrophages (si-SCR). Data obtained from the analysis of macrophages from ≥ 4 donors are presented. * (*p* < 0.05), ** (*p* < 0.005). *p*-values were calculated by a paired two-tailed t-test.

**Table 1 cells-09-00418-t001:** Pathways enriched among the genes positively correlated to ID4 in both the E-GEOD-18295 dataset and in basal-like breast cancers of the TCGA cohort. Analysis was performed using ConsensusPathDB, interrogating the KEGG pathway database (cutoff, *n* = 8 genes).

*p*-Value	Pathway	ID4-Correlated Genes
3.44 ×10**^−08^**	ECM-receptor interaction	*THBS2; LAMA4; ITGA1; COL9A2; COL9A3; AGRN; ITGA10; COL4A2; COL4A1; LAMB1; COL2A1*
1.39 × 10**^−06^**	PI3K-Akt signaling pathway	*COL4A1; ERBB3; PDGFD; PIK3R1; GHR; COL9A2; LAMB1; THBS2; COL9A3; CHRM1; GNG7; ITGA10; COL4A2; MAGI1; ITGA1; FGF7; LAMA4; COL2A1; LPAR4*
1.47 × 10**^−06^**	Amoebiasis	*PLCB1; TGFB2; PLCB4; PIK3R1; ACTN4; COL4A2; COL4A1; GNA11; LAMB1; LAMA4*
1.68 × 10**^−06^**	Focal adhesion	*THBS2; LAMA4; PIK3R1; ACTN4; ITGA1; COL9A2; COL9A3; ITGA10; COL4A2; COL4A1; LAMB1; PDGFD; COL2A1; MYL9*
4.92 × 10**^−05^**	Human papillomavirus infection	*DLG1; COL4A1; THBS2; LAMA4; MAGI1; FZD6; ITGA1; COL9A2; COL9A3; ITGA10; COL4A2; PIK3R1; FZD9; LAMB1; COL2A1; HEY1*
5.82 × 10**^−05^**	Protein digestion and absorption	*ELN; COL9A2; COL9A3; ATP1B1; COL5A2; COL4A2; COL4A1; COL2A1*
8.20 × 10**^−05^**	Vascular smooth muscle contraction	*PLCB1; NPR2; PLCB4; ARHGEF12; GNA11; EDNRA; PPP1R14A; CALD1; MYL9*
8.48 × 10**^−05^**	Regulation of actin cytoskeleton	*ARHGEF12; PDGFD; ACTN4; LIMK2; ITGA1; CHRM3; CHRM1; ITGA10; PIK3R1; FGF7; LPAR4; MYL9*
3.3 × 10**^−04^**	Pathways in cancer	*PLCB1; ARHGEF12; BMP4; PLCB4; COL4A1; FZD6; FZD9; LAMB1; TGFB2; GNG7; JUP; COL4A2; PIK3R1; GNA11; FGF7; LAMA4; EDNRA; LPAR4; HEY1*
4.8 × 10**^−04^**	Hippo signaling pathway	*DLG1; TGFB2; BMP4; FZD6; FZD9; CTGF; SNAI2; FRMD6; YAP1*
3.09 × 10**^−03^**	cGMP-PKG signaling pathway	*PLCB1; NPR2; PLCB4; SRF; ATP1B1; GNA11; EDNRA; MYL9*

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
