# Peer review of "Paracrine Signaling from Breast Cancer Cells Causes Activation of ID4 Expression in Tumor-Associated Macrophages"

_cells, 2020, doi:10.3390/cells9020418_

Round 1

Reviewer 1 Report

The authors prsent an interesting concept of breast cancer cell - TAM crosstal where expression of ID4 in BC cells induces expression of the same factor in TAMs in paracrine fashion. The ID4 expression in model macropahges was linked to several genes of interest in tumor biology - such as those involved in ECM modeling. 

The manuscript has general interest on the field and the data presented justifies the conclusions made. There were, however, a couple of minor issues to be resolved before publication of this work

Figure 5A, in legend only two panels are described (CTGF vs ID4 and YAP1 vs ID4). Why there is nothing told about the lowest CYR61 vs ID4 panel? Legend for C corrensponds to figures in 5B and again text for the CyR61/ID4 correlation is missing. Oddly legend text 5B apperas after legend "5C", but describes what can be seen as figure 5C.  It appears that this is a scrambled carryover of an old version of the figure legend...

Typo at Lane 150: "immunoprecipiatated" -> "immunoprecipitated"

Reviewer 2 Report

The study by Donzelli, et. al. provides an interesting story about TAMs reprogramming by ID4 in BC TMC. However, the current data is still premature for generating a significant finding. Authors need to further analyze the function of ID4 in TAMs to cancer progression by in vivo and in vitro assay. Moreover, they also need to address the mechanism how ID4 expression in tumor cells affects ID4 expression and TAMs reprogramming. Furthermore, whether blocking the potential route from tumor cells could reverse TAMs reprogramming and tumor malignancy. Additionally, current data are not even qualified for publication: western blotting data need to be provided in Fig. 1 and Fig. 2A and 2B; in Fig. 3A, ID4-HA overexpression was not observed in U937 cells.  

Reviewer 3 Report

The authors found that condition medium of breast cancer cells could induce ID4 expression in TAMs in vitro, and tried to analyze the underlying mechanism. However, there’s much to be evaluated and the manuscript needs to be further edited.

1, For figure 1A, did you compare RPMI FBS 10% HI-FBS with the CM? And as the data shows, CM of MB-468 induced robust higher level of ID4 in PBDM, did you test the ID4 expression in different tumor cell lines? Is there any correlation of ID4 expression in tumor cells and CM induced ID4 in PBDM?

2, Is there any association of ID4 and CCR2 of TAMs?

3, In figure2, the data shows that ID4 depletion didn’t reduce the ID4 level in PBDM to the level of RPMI treated group. Authors should include the reason in the discussion. Also, for the panel A, and B in figure2, did you detect protein level of ID4 expression using western blotting?

4, Did you investigate the transcription factors that induce ID4 expression of CM treated RAW264.7 cells?

Round 2

Reviewer 2 Report

The revised version has been improved and is sufficient for publication.

Reviewer 3 Report

The MS was improved and can be accepted.